# Assessment of ICAM-1 N-glycoforms in mouse and human models of endothelial dysfunction

Kellie Regal-McDonald[1,2], Maheshika Somarathna[3], Timmy Lee[3], Silvio H. Litovsky[1], Jarrod Barnes[4], J. M. Peretik[5], J. G. Traylor, Jr.[5], A. Wayne Orr[5], Rakesh P. Patel[1,2]*

**1** Department of Pathology, University of Alabama at Birmingham, Birmingham, Alabama, United States of America, **2** Center for Free Radical Biology, University of Alabama at Birmingham, Birmingham, Alabama, United States of America, **3** Division of Nephrology, Department of Medicine, University of Alabama at Birmingham, Birmingham, Alabama, United States of America, **4** Division of Pulmonary, Allergy, and Critical Care Medicine, Department of Medicine, University of Alabama at Birmingham, Birmingham, Alabama, United States of America, **5** Department of Pathology and Translational Pathobiology, Louisiana State University Health Sciences Center, Shreveport, Louisiana, United States of America

* rakeshpatel@uabmc.edu

**Data Availability Statement:** Relevant data are within the manuscript.

**Funding:** Funding for this study was received from NIH HL007918 (to KRM), NIH HL098435 (to

## Abstract

Endothelial dysfunction is a critical event in vascular inflammation characterized, in part, by elevated surface expression of adhesion molecules such as intercellular adhesion molecule-1 (ICAM-1). ICAM-1 is heavily N-glycosylated, and like other surface proteins, it is largely presumed that fully processed, complex N-glycoforms are dominant. However, our recent studies suggest that hypoglycosylated or high mannose (HM)-ICAM-1 N-glycoforms are also expressed on the cell surface during endothelial dysfunction, and have higher affinity for monocyte adhesion and regulate outside-in endothelial signaling by different mechanisms. Whether different ICAM-1 N-glycoforms are expressed *in vivo* during disease is unknown. In this study, using the proximity ligation assay, we assessed the relative formation of high mannose, hybrid and complex α-2,6-sialylated N-glycoforms of ICAM-1 in human and mouse models of atherosclerosis, as well as in arteriovenous fistulas (AVF) of patients on hemodialysis. Our data demonstrates that ICAM-1 harboring HM or hybrid epitopes as well as ICAM-1 bearing α-2,6-sialylated epitopes are present in human and mouse atherosclerotic lesions. Further, HM-ICAM-1 positively associated with increased macrophage burden in lesions as assessed by CD68 staining, whereas α-2,6-sialylated ICAM-1 did not. Finally, both HM and α-2,6-sialylated ICAM-1 N-glycoforms were present in hemodialysis patients who had AVF maturation failure compared to successful AVF maturation. Collectively, these data provide evidence that HM- ICAM-1 N-glycoforms are present *in vivo*, and at levels similar to complex α-2,6-sialylated ICAM-1 underscoring the need to better understand their roles in modulating vascular inflammation.

## Introduction

Inflammation is a carefully orchestrated response involving the release of pro-inflammatory factors and homing of immune cells to injured tissue. The vascular endothelium is a key player in the inflammatory process, responding to stressors such as oscillatory blood flow (that occurs

AWO), NIHHL133497 (to AWO), NIH DK109789 (to TL), NIH R00HL131866 (to JB), US Department of Veterans Affairs (to TL) and UAB Department of Pathology (internal) funds (to RPP). The funders had no role in study design, data collection and analysis, decision to publish, or preparation of the manuscript.

**Competing interests:** Dr. Lee is a consultant for Proteon Therapeutics, Merck, and Boston Scientific. All other authors have declared that no competing interests exist. This does not alter our adherence to PLOS ONE policies on sharing data and materials.

at bifurcations in the vessel) and increased pro-inflammatory factors [1–3]. The resulting activated endothelium provides a pro-adhesive surface that allows circulating immune cells to adhere and migrate into the inflamed tissue in a multistep process of capture, rolling, firm adhesion, and transmigration. These steps are mediated by multiple endothelial surface adhesion molecules [4–9].

Many surface and secreted proteins, including adhesion molecules, are N-glycosylated. While this post (or co-)-translational modification regulates protein stability and transport to the cell surface, these glycan structures are also important in mediating binding between cognate receptors, which is exemplified by binding between selectins and their sialyl Lewis$^X$ ligands [10, 11]. Relatively little is known, however, about how N-glycosylation of endothelial adhesion molecules may be regulated during inflammation [12]. Protein N-glycosylation occurs in the endoplasmic reticulum-Golgi network via a multi-step process involving initial attachment of a core oligosaccharide, comprised of $Glu_3Man_9GlcNAc_2$, onto the amide residue of asparagine within the consensus sequence N-X-S/T (where X cannot be proline). Sugars from this core are temporally and sequentially cleaved by ER/Golgi resident glycosidases, leading to high mannose structures, followed by partial rebuilding to hybrid N-glycan structures (collectively referred herein as hypoglycosylated N-glycans) by glycosyltransferases; after which multiple fucose, GlcNAc, and sialic acid residues are added to form complex N-glycoproteins [13]. While it is assumed that most glycoproteins require complete processing to the complex form for cell-surface expression, we and others have demonstrated that during atherogenic inflammation, the endothelial surface is characterized by decreased N-glycan complexity, or an increase in hypoglycosylated N-glycans [11, 14–16]. However, the exact endothelial proteins that harbor these hypoglycosylated N-glycans and their presence *in vivo* remains unclear.

Intercellular adhesion molecule-1 (ICAM-1) is an adhesion molecule that mediates leukocyte adhesion and whose expression is induced by pro-inflammatory stimuli. ICAM-1 has 8 putative N-glycosylation sites and we have previously shown that, in activated cultured endothelial cells, both high mannose/hybrid and complex (specifically α2,6-sialylated) ICAM-1 can be expressed on the cell-surface [15]. Interestingly, we observed that hypoglycosylated ICAM-1 N-glycoforms bound monocytes with higher affinity and regulated "outside-in" signaling via different mechanisms compared to ICAM-1 glycoforms containing complex, α-2,6-sialylated N-glycans. Furthermore, non-classical pro-inflammatory (CD14+CD16+) monocytes, but not classical (CD14+CD16-) monocytes, selectively recognize HM-ICAM-1 to roll and adhere to activated endothelial cells [15, 17, 18]. These findings suggest unique and distinctive functions for the different ICAM-1 N-glycoforms. However, whether these distinct ICAM-1 N-glycoforms are present *in vivo* is not yet known.

Herein, we tested whether different ICAM-1 N-glycoforms are present *in vivo* by modifying the DuoLink® proximity ligation assay (PLA) to detect co-localization of specific N-glycan structures and ICAM-1 in two disease states where endothelial dysfunction has been demonstrated to play a prominent role (see **Fig 1** for assay schematic and lectin specificity). Both human and mouse atherosclerotic tissue, as well as arteriovenous fistulas (AVF) of advanced chronic kidney disease patients, were evaluated for the presence of ICAM-1 N-glycoforms. Our data demonstrate that HM-ICAM-1 N-glycoforms are present in advanced disease states at similar, if not higher, levels compared to α-2,6-sialylated N-glycoforms. Importantly, HM-ICAM-1, but not α-2,6-sialylated ICAM-1 positively correlated with macrophage burden as assessed by CD68 staining suggesting a significant role for hypoglycosylated N-glycans in vascular inflammation.

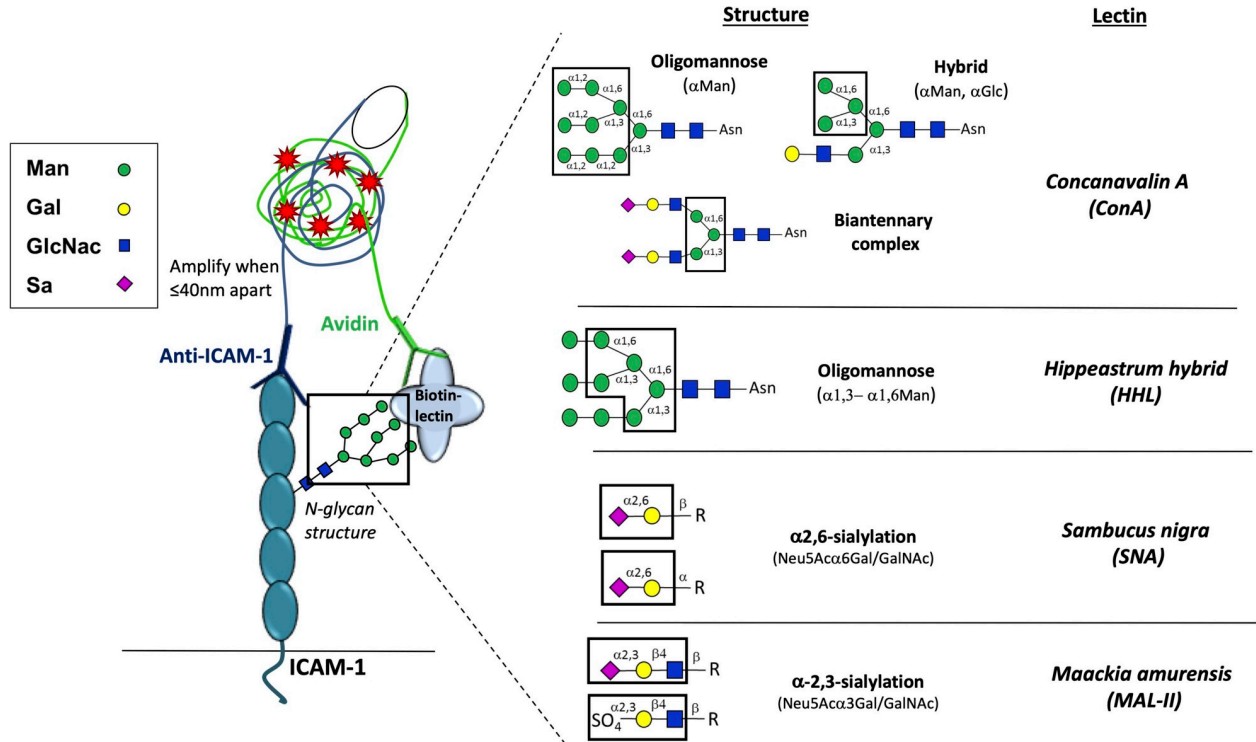

**Fig 1. Proximity-ligation assay (PLA) schematic and lectin specificity.** Biotinylated lectins with indicated specificities were added to first label specific sugars. Then, anti-ICAM-1 antibody and avidin, conjugated to complementary oligos, were added. When lectin-recognized sugar epitopes are less than 40 nm from anti-ICAM-1, the complementary oligos hybridize and amplify, producing red fluorescent puncta. Right panel shows N-glycan structures recognized by different lectins used (adapted from [19]). Man = Mannose; Gal = Galactose; GlcNac = N-acetylglucosamine; Sa = Sialic acid; R = varying N-glycan structures.

## Materials and methods

### Materials

DuoLink® proximity ligation assay kit was purchased from ThermoFisher (Waltham, MA). HistoPrep™ was purchased from Fisher Scientific (Hampton, NH). Anti-human ICAM-1 antibody was purchased from ThermoFisher (BMS108), and anti-mouse ICAM-1 antibody was purchased from Abcam (25375) (Cambridge, UK). *Concanavalin A* (ConA), *Sambucus Nigra* (SNA), *Hippeastrum Hybrid Amaryllis* (HHL), and *Maackia Amurensis* Lectin II (MAL-II) lectins were purchased from Vector laboratories (Birlingame, CA). Anti-ICAM-1 and avidin were conjugated to oligomer probes using the DuoLink Probemaker kits (Sigma) per manufacturer provided protocols. All other reagents were purchased from Sigma-Aldrich (St. Louis, MO) unless otherwise noted.

### Human vessel collection and processing

Human specimens were collected at the University of Alabama at Birmingham or Louisiana State University Health Sciences Center according to procedures and protocols approved by the University of Alabama at Birmingham Institutional Review Board and Louisiana State University Health Sciences Center Institutional Review Board respectively. Human arteries were obtained post-mortem, after authorization of autopsy that includes permission to remove tissues for research purposes. Vessel segments were fixed using formalin (10%) for 24h and then embedded in paraffin. Lesion type was determined according to the Stary scale from 1–5

[20] by a cardiovascular pathologist using the defining characteristics of fibrosis, calcification, and fatty deposits. All procedures were performed per Institutional Review Board approved protocols.

In addition, human vein samples were collected from subjects at the time of 1st and 2nd stage basilic vein transposition surgery. At the time of both surgeries, a circumferential piece of vein (~10–15 mm and adjacent to site of anastomosis creation) was excised and immediately stored in formalin for histology and immunohistochemistry (IHC) studies. Each venous tissue sample was embedded in paraffin as previously described [21]. Following paraffin embedding, each piece was sliced into 5μM sections for histological studies. These vein samples were collected under approval by the University of Alabama at Birmingham Institutional Review Board. Donor specimens from cadaveric donors at the time of organ harvesting were obtained using services from the National Disease Research Interchange, Philadelphia, PA, and fixed in formalin, as previously described [22].

## Mouse vessel collection and processing

8–10 week old male ApoE-/- mice were fed a high fat, Western-type diet (21% fat by weight, 0.15% cholesterol, and 19.5% casein without sodium cholate) for 12 weeks, and the innominate artery (i.e. brachiocephalic) and left carotid arteries collected for histological analysis. Both lesion and non-lesion areas of the same segment were analyzed and lesions identified by the presence of increased neointimal hyperplasia compared to non-lesion segments. Carotid arteries were also collected from male ApoE-/- mice fed a control diet. In the second model, 8–10 week old, male ApoE-/- mice underwent partial ligation of the left carotid artery to induce disturbed flow as described previously [19, 20]. After induction of anesthesia (5% isoflurane, followed by 2% isoflurane for maintenance of anesthesia), the external, internal, and occipital arteries were ligated with a 7–0 silk suture leaving the superior thyroid artery patent, and the resulting disturbed flow pattern was verified by ultrasound measurements using a VisualSonics VEVO3100 system. When this model is performed in atherosclerosis-prone ApoE-/- mice, accelerated atherosclerosis develops in the left carotid artery due to disturbed flow compared to the paired right carotid artery [21]. Mice were euthanized by pneumothorax under isoflurane anesthesia, and then ex-sanguinated followed by vessel collection. For all samples, the adventitia was carefully removed and vessel segments fixed in formalin (10%) for 24h and then embedded in paraffin blocks for mounting to slides. All procedures were performed according to LSU Health Sciences Center-Shreveport and University of Alabama at Birmingham IACUC approved protocols.

## Tissue immunofluorescence

Paraffin-embedded serial tissue sections were rehydrated using standard protocols. Briefly, slides were treated with xylene followed by decreasing Histoprep concentrations (100%, 95%, 85%, 75%) before immersion in 1x PBS. Antigen retrieval was accomplished using 10mM sodium citrate buffer (pH 6.0) and heated for 12 minutes in the microwave. Following antigen retrieval, tissues were washed in PBS and proximity-ligation, total ICAM-1, or CD68 staining performed as outlined below.

## Proximity-ligation assay

Briefly, 20 μg of the following biotinylated lectins: HM and hybrid N-glycan-specific lectin ConA, the α2,6-sialyation specific lectin SNA, the HM-specific lectin HHL, or the α2,3-sialylation specific lectin MAL-II were added for 30 minutes at 4°C (see **Table 1**). These lectins were chosen to provide broad coverage of different N-glcyan structures. After washing twice with

**Table 1. Atherosclerosis patient demographics (N.D. = no data).**

| Patient # | Age | Sex and Race | Atherosclerotic Lesion Type | Vessel Sample(s) |
|---|---|---|---|---|
| 1 | 28 | Female, White | 1<br>2 | Right coronary<br>Circumflex |
| 2 | 47 | Male, White | 2 | Right coronary |
| 3 | 22 | Male, N.D. | 3 | Left anterior descending |
| 4 | 22 | Male, African American | 3 | Right coronary |
| 5 | 66 | Male, African American | 4<br>5<br>5 | Circumflex<br>Left anterior descending<br>Right coronary |
| 6 | 58 | Male, White | 2 | Thoracic Aorta |
| 7 | 31 | Male, White | 1 | Thoracic Aorta |
| 8 | 64 | Female, White | 5<br>3 | Left coronary<br>Circumflex |
| 9 | 45 | Female, White | 2 | Aorta |
| 10 | 55 | Female, White | 2 | Left anterior descending |
| 11 | 58 | Male, African American | 4<br>3 | Left anterior descending<br>Left coronary |
| 12 | 78 | Male, White | 3 | Right coronary |

PBS, samples were blocked with 1X Carbo-Free blocking solution (Vector Labs) for 30 min at 20–25˚C. Immediately following blocking, samples were incubated with oligo-tagged avidin or anti-ICAM-1 (10 μg/mL) for one hour at 37˚C followed by ligation and amplification steps as per the protocol. Both anti-mouse and human ICAM-1 antibodies recognize extracellular (domain 1) epitopes on ICAM-1. Slides were left to dry and mounted using the DuoLink® mounting medium **containing DAPI.**

## Total ICAM-1 and CD68 immunofluorescence

Slides were processed as described above. After blocking, slides were incubated with mouse or human anti-ICAM-1 or CD68 O/N at 4˚C, followed by a 1 hour incubation at 20–25˚C with the species-appropriate secondary antibody conjugated to Alexa Fluor 594. After washing, slides were left to dry and mounted using DuoLink® mounting medium containing DAPI.

## Image acquisition and analysis

Images were acquired on a BioTek Lionheart live cell imager using DAPI (377/447), GFP (469/525), and Texas Red (586/647) filters. LED intensity, camera gain, and integration time for each channel was uniform across all samples. Images at three random locations within lesion and non-lesion areas were recorded per section. Images were collected at 4x and 60x magnification. Image quantification was performed using either Gen5 (Biotek) or ImageJ software (NIH). To quantify PLA images, 3 random fields per group, per experiment were selected and number of puncta counted using ImageJ particle analysis software (NIH).

## Statistics

All statistical analysis was performed using GraphPad Prism software. Paired student t-tests were utilized in comparing the different areas (lesion vs. non-lesion) from the same vessel or left and right carotids from the same animal. Unpaired t-test was used to compare human vessels with different lesion stages. A p-value less than 0.05 was considered significant.

## Results

### Hypoglycosylated ICAM-1 is present in mouse atherosclerotic lesions

Vessels from ApoE-/- mice were collected after 12 weeks feeding of a control or high fat diet, with the latter being an established method to reproducibly promote atherosclerosis [7, 22, 23]. Total ICAM-1 levels (measured using an anti-ICAM-1 antibody that binds to epitopes on the N-glycan-devoid domain 1 of the protein) were detected on the luminal surface of lesions, consistent with prior reports of up-regulation of this adhesion molecule on the endothelium [24–26] (**Fig 2A**). No ICAM-1 was detected on areas of the vessel devoid of lesions, nor on vessels collected from mice fed a normal chow diet. To determine which N-glycoforms of ICAM-1 were present, we used the PLA assay (**Fig 1**), where red puncta indicate positive staining that the two epitopes are within 40nm of each other. To measure HM and hybrid ICAM-1, samples were labeled with biotinylated ConA, followed by treatment with avidin and an anti-ICAM-1 antibody both tagged with complementary oligonucleotides. High mannose N-glycans, α-2,6-sialylation, and α-2,3-sialylation on ICAM-1 were measured using the same method with HHL, SNA and MAL-II lectins, respectively. **Fig 2A** shows red puncta for the ICAM-1 N-glycoforms in lesion areas only; no PLA staining was observed in non-lesion areas of the same vessels nor in vessels from ApoE-/- mice fed normal chow (control diet). As an immunostaining control, the antibody against ICAM-1 or avidin were excluded (**Fig 2B and 2C** respectively) and showed no PLA-positive staining associated with lesions. **Fig 2D and 2E** shows that total ICAM-1 and HM / hybrid N-glycoforms of ICAM-1 were significantly increased in lesion vs. non-lesion regions, with similar trends with α-2,6-sialylated ICAM-1 noted (p<0.07). Macrophage burden was assessed by CD68 staining in lesion and non-lesion regions; these measurements were performed on serial sections from the same preparations used for proximity ligation assay staining. CD68 staining was significantly higher in lesion areas compared to non-lesion areas (**Fig 2F and 2G**). *Note*: Since different lectins will bind their target epitopes with distinct affinities, direct comparison between lectins is not possible.

We also collected vessels from ApoE-/- mice that underwent partial carotid ligation, which creates oscillatory flow in the left carotid artery while leaving the right carotid artery as an internal control. This model rapidly causes endothelial dysfunction and atherosclerosis [19]. Both the left and right carotid arteries from mice that underwent partial ligation were collected 7 days post-ligation and stained for total ICAM-1 and ICAM-1 N-glycoforms. **Fig 3A–3C** show representative images and quantification of ICAM-1 and ICAM-1 N-glycoform staining. Total ICAM-1, HM / hybrid N-glycoforms were increased in the ligated left carotid artery (LC) compared to the control right carotid artery (RC). CD68 staining was also significantly higher in the LC compared to the control RC (**Fig 3D and 3E**).

**Fig 4A–4C** show correlations between CD68 staining and HM / hybrid, HM, and α-2,6-sialylated, ICAM-1 respectively, from ApoE-/- mice fed high fat diets and **Fig 4D–4F** after partial carotid ligation. Significant positive correlations were observed between CD68 and HM / hybrid and HM-ICAM-1 in both mouse models, but only HFD-fed mice had a significant positive correlation between CD68 and α-2,6-sialylated, ICAM-1.

### HM / hybrid and α2,6-sialylated ICAM-1 are present in advanced human atherosclerotic lesions

To determine whether distinct ICAM-1 N-glycoforms are present in human atherosclerosis and whether they were dependent on disease severity, human vessels were collected at autopsy and luminal surface total ICAM-1 and HM / hybrid, α2,3-sialylated, and α2,6-sialylated ICAM-1 were measured via PLA. **Table 1** summarizes patient demographics, vessel location,

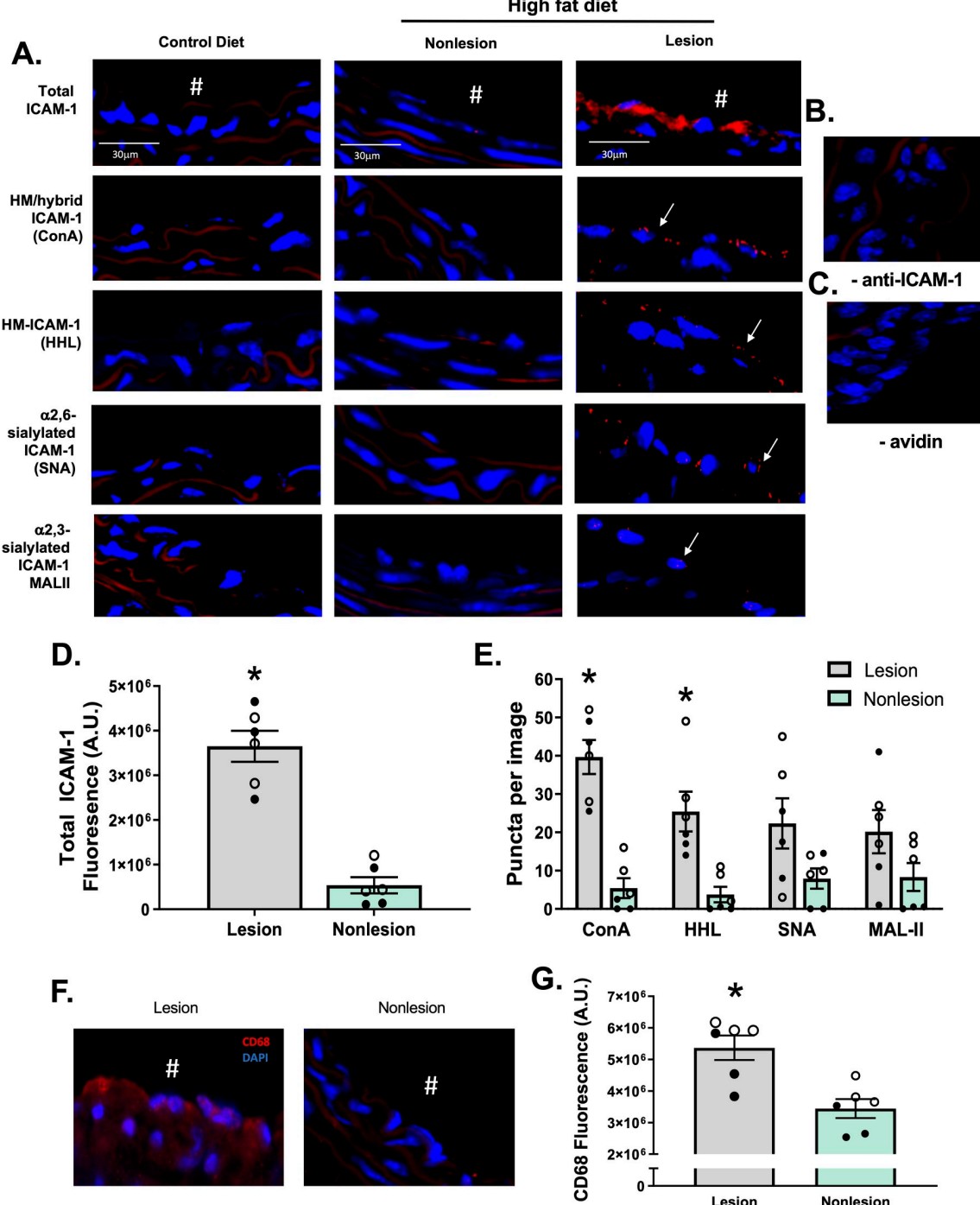

**Fig 2. HM epitopes co-localize with ICAM-1 in high fat-induced mouse atherosclerosis.** Total, HM / hybrid, α-2,6-sialylated, and α-2,3-sialylated ICAM-1 were measured in the innominate and left carotid arteries from ApoE-/- mice fed a normal or high fat diet. **A)** Shown are representative images of innominate arteries from paired lesion and non-lesion areas of the same vessel section. Red staining represents total ICAM-1, red puncta represent positive PLA staining for specific ICAM-1 N-glycoforms (indicated by arrows), and blue staining represents DAPI. # indicates the lumen of each vessel. Panels **B and C** show PLA staining of lesion areas when the anti-ICAM-1 antibody or avidin were excluded. **Panel D** shows total ICAM-1 staining in lesion versus non-lesion areas. *p<0.05 by t-test. **Panel E** shows number of puncta for HM / hybrid, HM, α-2,6-sialylated, and α-2,3-sialylated ICAM-1 in lesion versus non-lesion areas. Data are mean ± SEM, each symbol represents a distinct mouse, n = 6. *p<0.05 compared to non-lesion via paired t-test. **F.** Representative images of CD68 staining (red) in lesion vs. non-lesion areas. **G.** Quantitation of CD68 staining in lesion and non-lesion areas. Data are mean ± SEM, each symbol represents a distinct mouse, n = 6. * = p<0.05 compared to non-lesion via paired t-test. ● from innominate arteries and ○ from left carotid arteries.

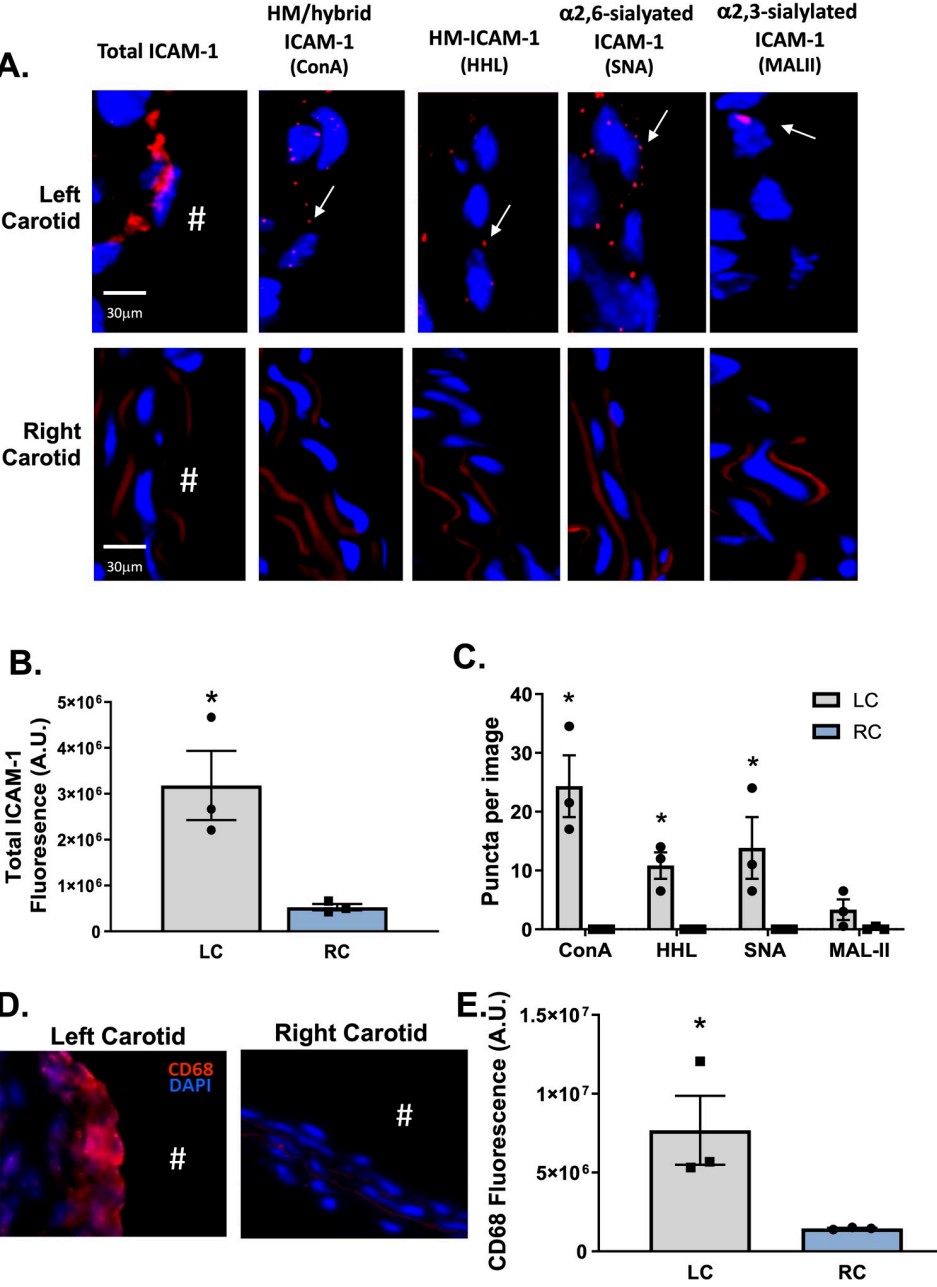

**Fig 3. HM / hybrid, HM, and α-2,6-sialylated ICAM-1 are increased in mouse atherosclerosis after induction of disturbed flow *in vivo*. Panel A** shows representative images of total ICAM-1 HM / hybrid, HM, α2,6-sialylated, and α2,3-sialylated ICAM-1 in the left carotid artery (after partial ligation) and paired right carotid artery (control). Positive PLA puncta are indicated by arrows. **Panels B & C** show total ICAM-1 staining in left versus right carotid artery and ICAM-1 N-glycoforms puncta, respectively. Data are mean ± SEM, each symbol represents an individual mouse, n = 3. *p<0.05 compared to RC via paired t-test. **D.** Representative images of CD68 staining (red) in LC vs. RC areas in mice. **E.** Quantitation of CD68 staining in LC and RC areas. Data are mean ± SEM, n = 3. *p<0.05 compared to RC via t-test. # indicates the lumen of each vessel.

and lesion type. ICAM-1 expression increased as a function of disease severity with staining visible in type 3 and 5 lesions (**Fig 5A and 5B**). Similarly, HM-/ hybrid, α-2,6-sialylation, and α-2,3-sialylation N-glycoforms were only detected in advanced lesions (**Fig 5A and 5C**) – **Fig 5D**

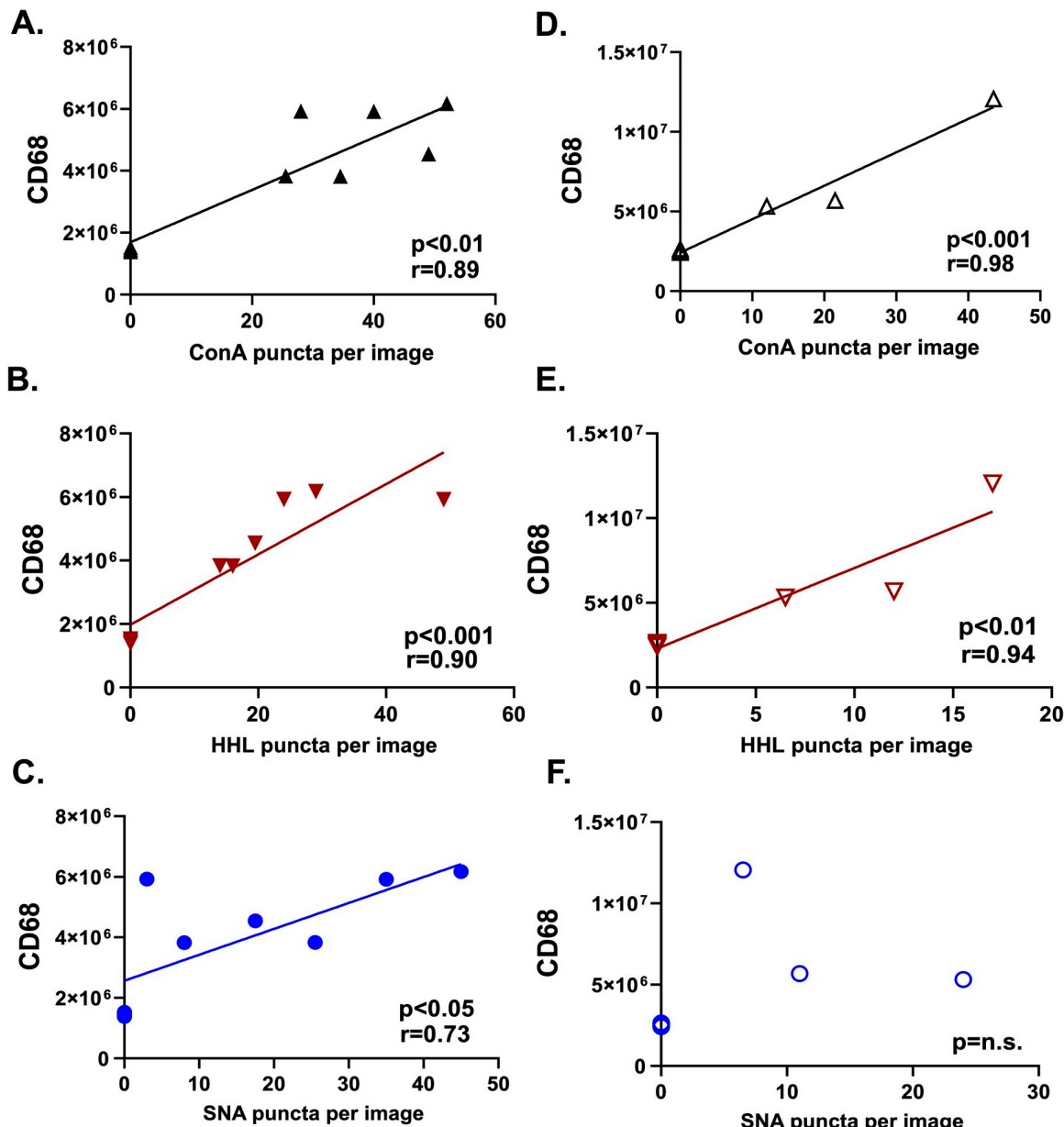

**Fig 4. CD68 macrophage staining positively correlates with HM-ICAM-1 in animal models of atherosclerosis. A-C.** CD68 staining (fluorescence) as a function of ConA, HHL, and SNA puncta, respectively, for ApoE-/- mice fed HFD diet. n = 6 animals; n = 12 paired lesion and non-lesion vessel areas. **D-F**. CD68 staining (fluorescence) as a function of ConA, HHL, and SNA puncta, respectively, for ApoE-/- mice after partial carotid ligation n = 3 animals; n = 6 vessel samples. Best fit lines determined by linear regression with Pearson correlation analyses with indicated coefficients and p-values shown in each panel.

and **5E** show specificity for HM / hybrid and sialylated-ICAM-1 staining; no PLA-positive staining was observed when individual PLA reagents were omitted.

Finally, CD68 staining was significantly higher in advanced lesions compared to earlier stage lesions (**Fig 6A and 6B**) or compared to IgG control staining of advanced lesions. Further, CD68 staining had a significant positive correlation with HM-ICAM-1 levels with a trend towards significance (P<0.09) noted for correlation between α-2,6-sialyated ICAM-1 and CD68 levels. (**Fig 6C–6E**).

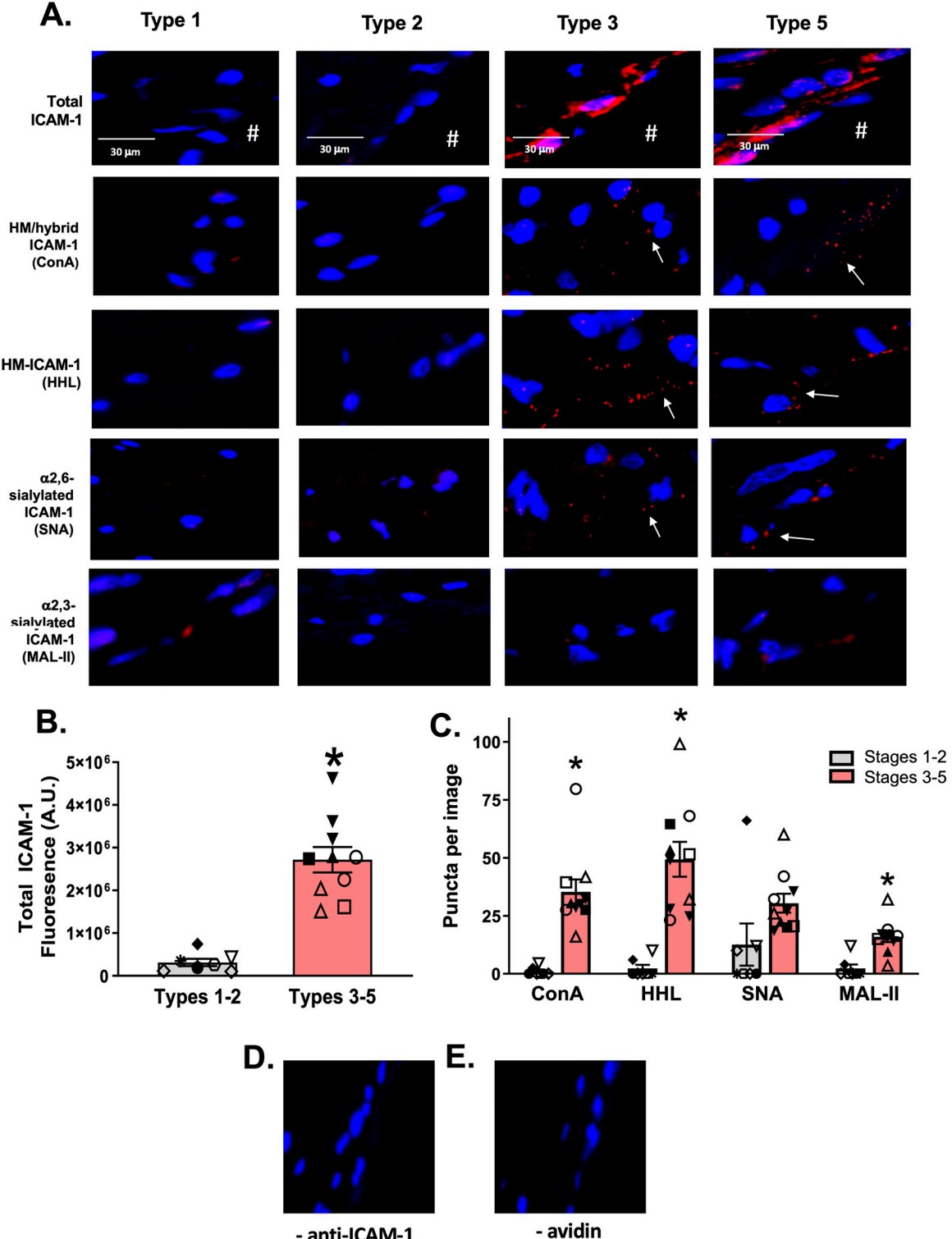

**Fig 5. HM / hybrid ICAM-1 is increased in human atherosclerosis. Panel A** shows representative images of total ICAM-1 (red staining) and specified N-glycoforms in human vessels with lesions spanning types 1–5. Red puncta represent positive PLA staining (as indicated by arrows). **Panel B** shows the quantification of total ICAM-1 in early (1–2) and late (3–5) disease stages. Each symbol represents a different patient, with same symbol representing multiple vessels from the same patient. Data are mean ± SEM, n = 7–10. * p<0.05 compared to types 1–2 via unpaired t-test. **Panel C** shows the quantification of HM / hybrid, HM, α-2,6-sialylated, and α-2,3-sialylated ICAM-1 puncta in early (1–2) and late (3–5) disease stages. * = p<0.05 compared to early stage lesions by t-test. Data are mean ± SEM. n = 7–10. **Panels D** and **E** show staining of lesion areas when the anti-ICAM-1 antibody or avidin were excluded.

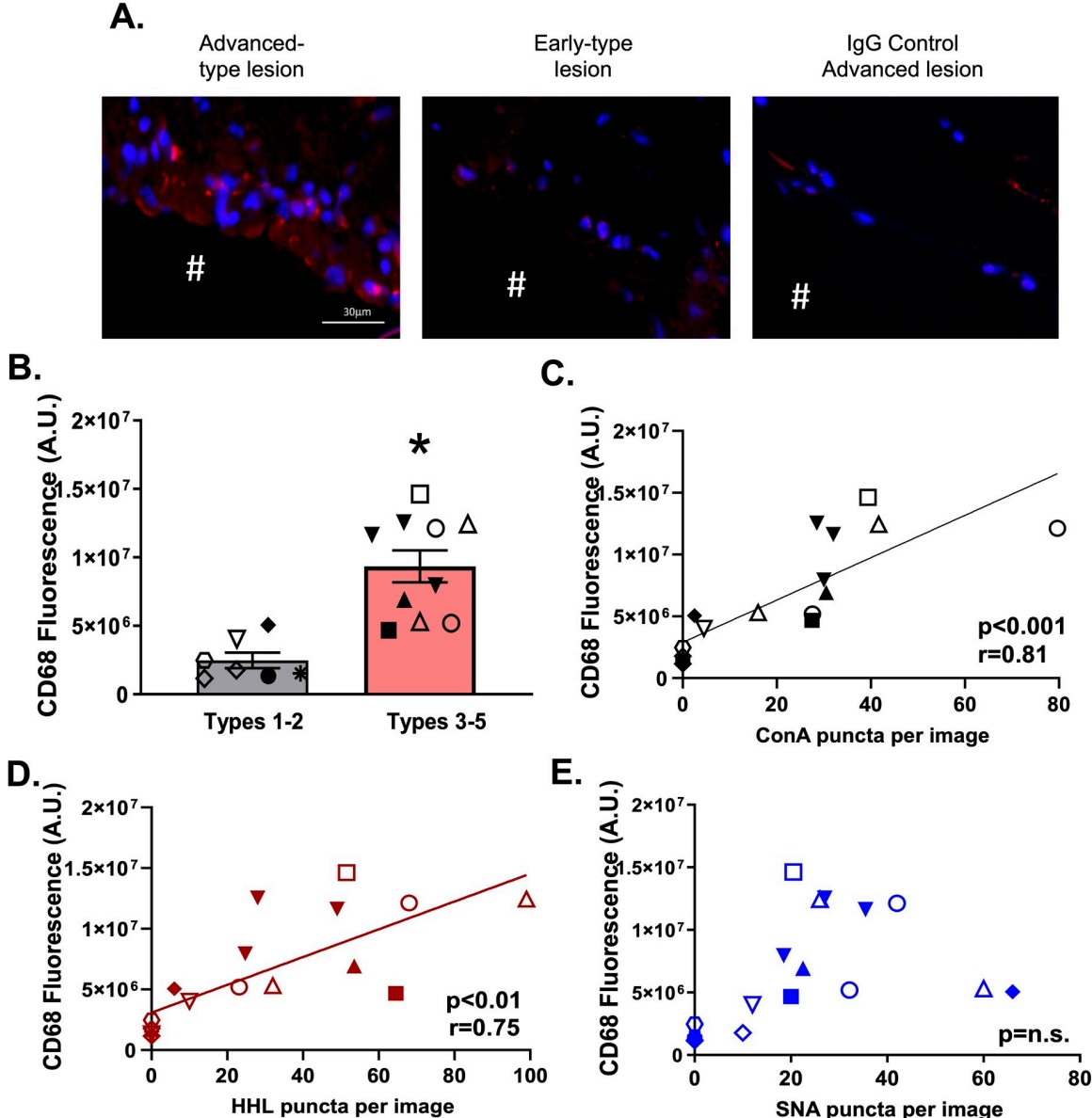

**Fig 6. CD68 macrophage staining positively correlates with HM / hybrid ICAM-1 in human atherosclerosis.** Panels **A** and **B** show representative images and quantitation, respectively, of CD68 staining in advanced and early type lesions, as well as IgG control staining. Data are mean ± SEM, n = 7–10 * p<0.05 compared to types 1–2 via unpaired t-test. **C-E.** ConA, HHL, and SNA puncta plotted against CD68 staining. Data are from n = 17 vessel samples collected from 12 subjects; each symbol represents a different subject. Best fit lines were determined by linear regression analyses indicated coefficients determined by Pearson correlation analyses.

## HM / hybrid and α2,6-sialylated ICAM-1 are present in patients on hemodialysis with failed AVFs

Advanced chronic kidney disease (CKD) patients that reach end-stage renal disease require hemodialysis. A common limitation with this therapy is failure of the AVF to successfully mature due to endothelial dysfunction and inflammation mediated in part by oscillatory flow [27, 28]. Basilic vein samples from CKD patients that had undergone surgical AVF creation were obtained and subjected to PLA for HM/hybrid, α-2,3-sialylated, and α-2,6-sialylated

ICAM-1. **Table 2** shows patient demographics, and the success vs. failure status of AVF. **Fig 7A** show representative images indicating that total ICAM-1, HM / hybrid, α-2,6-sialylated, and α-2,3-sialylated ICAM-1 were present in the failed AVFs. Little to no ICAM-1 was detected in successful AVFs (**Fig 7A and 7B**). HM / hybrid and α-2,6-sialylated ICAM-1 levels were significantly higher in failed AVFs compared to successful AVFs, with α-2,3-sialylated ICAM-1 also being elevated (p<0.06) (**Fig 7C**). No staining for ICAM-1 or associated N-glycoforms (only tested with ConA and SNA) were observed in control (no AVF) healthy donor veins (n = 3).

## Discussion

Inflammation is a carefully orchestrated process involving the homing of immune cells to inflamed tissues, a process mediated by adhesion molecules on the endothelial cell surface [5]. Surface adhesion molecules are heavily N-glycosylated with ~20–50% of their observed molecular weights attributed to N-glycans. While insights into the role of protein N-glycosylation in regulating inflammation have primarily focused on circulating immune cells (e.g. CD44 and other selectin ligands on neutrophils and T-cells [10,21,29], relatively little attention has been paid to the impact N-glycans may have on the function of endothelial adhesion molecules in diseases mediated by endothelial dysfunction.

ICAM-1 is a well-established mediator of increased monocyte-endothelial interactions and interestingly, compared to the related adhesion molecule, VCAM-1, contains more N-glycans as a proportion of its overall molecular weight (~50% in ICAM-1 vs. ~25% in VCAM-1) [30]. Numerous studies have shown that deletion of ICAM-1 or blocking its ability to engage Mac-1 or LFA-1 on leukocytes prevents inflammation [31–34]. It is also known that N-glycosylation can modulate ICAM-1 function. For example, Mac-1 binding is higher to ICAM-1 bearing HM structures[35]. Moreover, different ICAM-1 N-glycoforms may regulate tumor burden and inflammatory signaling, and that the degree of ICAM-1 N-glycosylation can change depending on the cell in which it is expressed [36–40]. In our previous work, we showed that HM / hybrid ICAM-1 was formed in activated endothelial cells, and this hypoglycosylated ICAM-1 N-glycoform mediated higher affinity binding to monocytes compared to fully processed, sialylated ICAM-1 N-glycoforms. In addition, ligation dependent interactions between ICAM-1 and the actin cytoskeleton was also distinctly regulated between HM / hybrid ICAM-1 and complex-ICAM-1[15,30].

While the aforementioned studies suggest altered functions for hypoglycosylated vs. complex sialylated ICAM-1, whether these N-glycoforms are expressed in disease is not known and important to establish translational significance. By using vessels from patients with varying stages of atherosclerotic lesions, from CKD patients with successful or failed AVFs, and from two atherosclerotic mouse models (ApoE-/- mice fed high-fat diet and ApoE-/- mice post-partial carotid ligation), we show that HM / hybrid and complex ICAM-1 N-glycoforms are found in vascular lesions but not in non-lesion areas. Interestingly, HM / hybrid ICAM-1 levels displayed stronger correlations with CD68 macrophage staining in human and mouse lesions, suggesting that hypoglycosylated ICAM-1 may play a more prominent role in mediating monocyte recruitment compared to α-2,6-sialylated ICAM-1. Consistent with this hypothesis, our previous data showed that the affinity of monocyte adhesion to HM / hybrid ICAM-1 was greater compared to α-2,6-sialylated ICAM-1[18,30].

To date, very few studies have focused on N-glycosylation of endothelial cells and their role in disease. The complexity of studying surface adhesion proteins and their N-glycan patterns may be a contributing factor to the lack of studies in this area. For example, ICAM-1 contains 8 putative N-glycosylation sites. This, coupled with the different possible combination of N-

**Table 2. AVF patient demographics.**

| Patient # | AVF Status | Age | Sex and Race | Vessel Sample(s) |
|---|---|---|---|---|
| 13 | Successful | 55 | Male, White | Basillic |
| 14 | Successful | 77 | Female, African American | Basillic |
| 15 | Successful | 54 | Male, White | Basillic |
| 16 | Successful | 72 | Female, African American | Basillic |
| 17 | Failed | 73 | Female, African American | Basillic |
| 18 | Failed | 63 | Female, African American | Basillic |
| 19 | Control–no AVF | 58 | Female, White | Basillic |
| 20 | Control–no AVF | 69 | Female, White | Basillic |
| 21 | Control–no AVF | 56 | Female, White | Basillic |
| 22 | Failed | 58 | Female, African American | Basillic |
| 23 | Failed | 36 | Male, African American | Basillic |

linked sugars ranging from HM- to complex N-glycans, means that ICAM-1 may exist in >200,000 possible N-glycoforms (calculated from data in Lau et al [41]. This complexity clearly makes identification of specific N-glycoforms challenging [17,42]. It is for this reason we employed 4 different lectins in this study to identify different ICAM-1 N-glycoforms. By using lectins that recognize high-mannose, hybrid, and two different types of sialic acid linkages, we detected ICAM-1 decorated with at least 4 different types of N-glycans *in vivo* (*Note*: ConA and HHL have some overlapping N-glycan binding specificities). To our knowledge, this data represents the first study demonstrating distinct N-glycoforms on ICAM-1 *in vivo*. We do note limitations of our study, namely the sole reliance on the PLA assay to discern HM, hybrid, and sialylated-ICAM-1 and that this assay only informs on spatial proximity; it is possible that HM-epitopes on other proteins and ICAM-1 are co-expressed. However, our previous data show that ICAM-1 immunoprecipitated from human atherosclerotic lesions has a similar molecular weight (75KDa) to HM / hybrid ICAM-1; complex ICAM-1 is close to 100KDa, supports conclusions that a HM-ICAM-1 N-glycoform is present *in vivo* [15]. An additional limitation is quantitation of PLA-positive puncta. Differential affinities for binding between specific N-glycan structures and the different lectins could lead to different staining intensities. For this reason, we quantified data based only on the number of positive puncta and not on staining intensity. However, we recognize that relative differences in lectin binding may have affected sensitivity for detecting discreet PLA-positive puncta. Potential limitations in detection also preclude definitive conclusions regarding temporal expression of ICAM-1 and its N-glycoforms as a function of disease. Type 2 lesions are characterized by lipid laden macrophages; however neither ICAM-1 protein, nor its N-glycoforms, were detectable in these lesions in our methods. ICAM-1 expression in early atherosclerosis has been studied, and in some cases has been found to be minimal in earlier stages of the disease [43,44], which may explain the lack of detection in our methods. Further, the role of ICAM-1 in early atherosclerosis is still debated [45], and our data suggests this HM-ICAM-1 may be important in later stages of the disease.

Further studies using more specific and selective approaches for N-glycan analyses are required to determine the exact N-glycoforms of ICAM-1 present in vascular inflammation and their function. Little is known about the regulation and control of different N-glycoforms, and if the regulation may be disease specific. For example, while our data showed little α-2,3-sialylated ICAM-1 in human atherosclerosis, it was present in failed AVFs from humans. This observation perhaps indicates that regulation of N-glycoforms depends on the disease state. Future studies in the lab will focus on the regulation of N-glycans during inflammation;

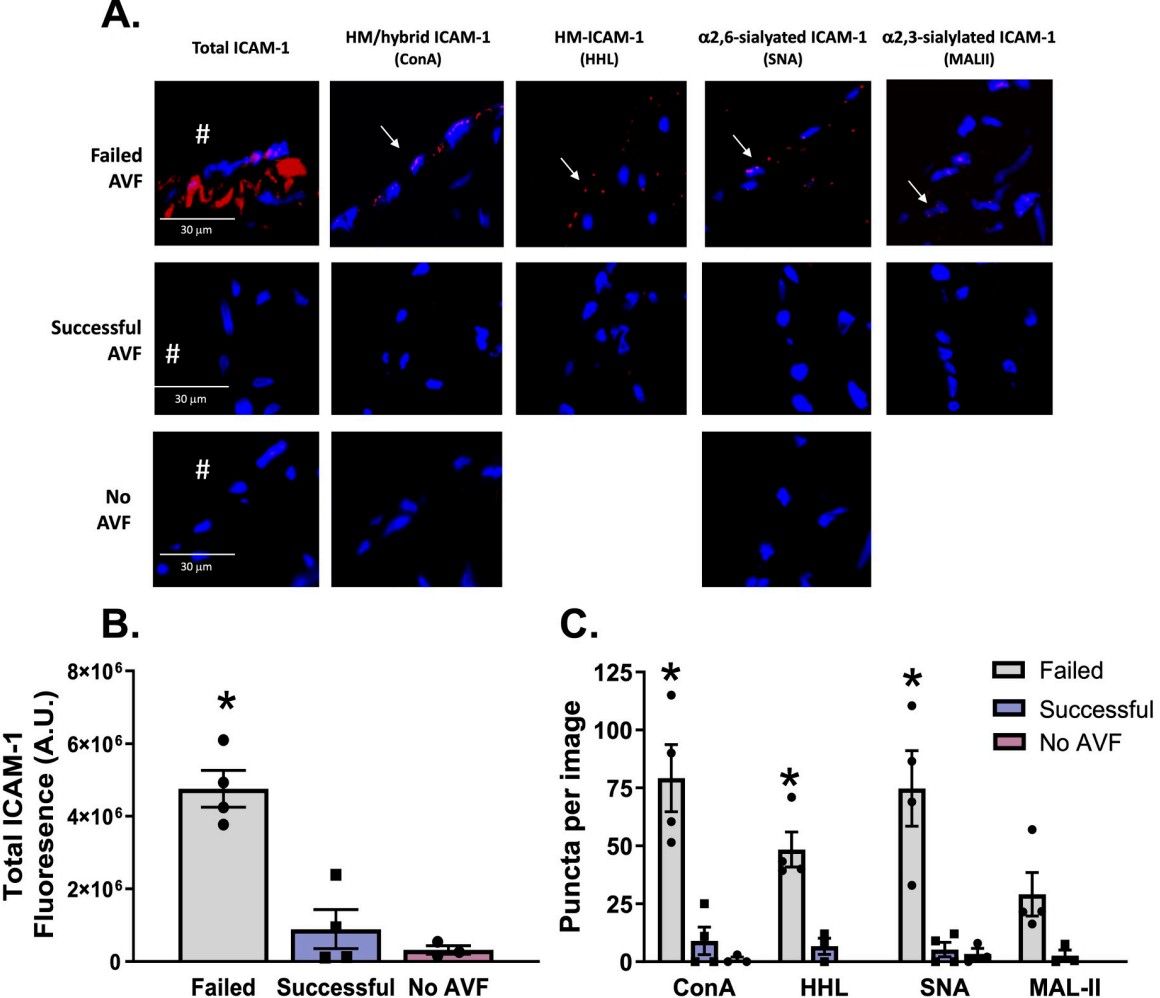

**Fig 7. HM / hybrid, HM, α-2,6-sialylated, ICAM-1 are increased CKD patients with failed arteriovenous fistulas. Panel A** shows total ICAM-1 (red staining) and specified N-glycoforms in vessels from CKD patients with failed or successful AVF creation. Red puncta represent positive PLA staining (as indicated by arrows). **B)** Quantification of total ICAM-1 signal in failed and successful AVF samples (n = 4 each). Error bars are mean ± SEM; * p<0.05 compared to successful AVF samples by t-test. **C)** Quantification of HM / hybrid, HM, α-2,6-sialylated, and α-2,3-sialylated ICAM-1 as puncta per total ICAM-1 signal. Error bars are mean ± SEM; * p<0.05 compared to successful AVF samples by unpaired t-test.

specifically by looking at the enzymes responsible for N-glycosylation, such as α-mannosidases and glycosyltransferases.

Distinct biological functions of ICAM-1 N-glycoforms may also have therapeutic implications. Attempts to therapeutically target ICAM-1 to abrogate atherosclerosis *in vivo* have been successful in animal models; for example, ICAM-1 deficiency and anti-ICAM-1 antibody treatment protects against atherosclerosis in ApoE-deficient mice [34,46]. However, anti-ICAM-1 therapies have not translated into humans. Treatment with functional blocking anti-ICAM-1 antibodies have had no effect in improving kidney allograft rejection rates nor mortality rates in stroke patients, and in fact induced a neutrophil-dependent pro-inflammatory response in stroke patients [47–49]. Moreover, a general concern with a long-term anti ICAM-1 therapeutic strategy is the potential for inhibiting innate immunity [50]. ICAM-1 is a diverse protein with functions spanning mitogenic signaling, leukocyte adhesion, and cell survival mechanisms [51–53], and therefore global targeting can result in a potentially detrimental

disruption of normal responses. Our findings that different N-glycoforms of ICAM-1 are expressed, coupled with these having distinct functions [15,18], may offer new therapeutic strategies that involve targeting hypoglycosylated HM-ICAM1 specifically. We posit that blocking HM / hybrid ICAM-1 may lead to selective attenuation of monocyte ingress into atheroprone endothelial beds. A better understanding of the glycosylation patterns of the activated endothelium throughout the course of disease may yield more selective therapeutics that target endothelial inflammation in a disease and vascular bed-specific manner.

## Author Contributions

**Conceptualization:** Kellie Regal-McDonald, Timmy Lee, Silvio H. Litovsky, Jarrod Barnes, Rakesh P. Patel.

**Data curation:** Kellie Regal-McDonald.

**Formal analysis:** Kellie Regal-McDonald.

**Funding acquisition:** Timmy Lee.

**Investigation:** Kellie Regal-McDonald.

**Methodology:** Timmy Lee, J. M. Peretik, A. Wayne Orr, Rakesh P. Patel.

**Project administration:** Rakesh P. Patel.

**Resources:** Maheshika Somarathna, Timmy Lee, Silvio H. Litovsky, Jarrod Barnes, J. M. Peretik, J. G. Traylor, Jr., A. Wayne Orr, Rakesh P. Patel.

**Supervision:** Rakesh P. Patel.

**Validation:** Kellie Regal-McDonald, Rakesh P. Patel.

**Visualization:** Kellie Regal-McDonald.

**Writing – original draft:** Kellie Regal-McDonald.

**Writing – review & editing:** Kellie Regal-McDonald, Maheshika Somarathna, Timmy Lee, Silvio H. Litovsky, Jarrod Barnes, J. M. Peretik, J. G. Traylor, Jr., A. Wayne Orr, Rakesh P. Patel.

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
