## [Decision Letter · Decision Letter 0]

29 Aug 2019

PONE-D-19-20764

Assessment of ICAM-1 N-Glycoforms in Mouse and Human Models of Endothelial Dysfunction

PLOS ONE

Dear Dr. Rakesh Patel,

Thank you for submitting your manuscript to PLOS ONE. After careful consideration, we feel that it has merit but does not fully meet PLOS ONE’s publication criteria as it currently stands. Therefore, we invite you to submit a revised version of the manuscript that addresses the points raised during the review process.

As you will gather from the reviews, the referees identified several methodological problems and questionable functional significance of the current findings.   The editor concurs.   Major issues include low resolution of images and lack of support data for PLA (e.g. western analysis).  Other concerns pointed by reviewers should also be addressed.   

We would appreciate receiving your revised manuscript within three months. To enhance the reproducibility of your results, we recommend that if applicable you deposit your laboratory protocols in protocols.io, where a protocol can be assigned its own identifier (DOI) such that it can be cited independently in the future. For instructions see: http://journals.plos.org/plosone/s/submission-guidelines#loc-laboratory-protocols

We look forward to receiving your revised manuscript.

Kind regards,

Tohru Fukai

Academic Editor

PLOS ONE

Journal Requirements:

2. Thank you for including your ethics statement: "Human specimens were collected according to University of Alabama at Birmingham Institutional Review Board and Louisiana State University Health Sciences Center Institutional Review Board.

All animal procedures were performed according to LSU Health Sciences Center-Shreveport IACUC approved protocols. Anesthesia was administered using isoflurane and euthanasia by pneumothorax under isoflurane anesthesia followed by ex-sanguination."

Please amend your current ethics statement to confirm that your named institutional review board or ethics committee specifically approved this study.

a) Please provide an amended Funding Statement that declares *all* the funding or sources of support received during this specific study (whether external or internal to your organization) as detailed online in our guide for authors at http://journals.plos.org/plosone/s/submit-now.  

b) Please state what role the funders took in the study.  If any authors received a salary from any of your funders, please state which authors and which funder. If the funders had no role, please state: "The funders had no role in study design, data collection and analysis, decision to publish, or preparation of the manuscript."

Dr. Lee is a consultant for Proteon Therapeutics, Merck, and Boston Scientific. All other authors have declared that no competing interests exist.

Reviewers' comments:

Reviewer's Responses to Questions

5. Review Comments to the Author

Reviewer #1: The study by Regal-McDonald et al used PLA to investigate co-localization of specific N-glycan structures and ICAM-1 in mouse and human atherosclerotic arteries and human AVF of chronic kidney disease patients (models of endothelial dysfunction). The results demonstrated that hypoglycosylated and α-2,6-sialylated ICAM-1 N-glycoforms are present in atherosclerotic murine and human arteries and AVF patients. The results also indicated that HM-ICAM-1 levels positively correlated with macrophage burden in both murine and human atherosclerotic tissue. The study is timely but lacks functional information on the role of ICAM-1 N-glycoforms in atherosclerosis or AVF.

1. The figures of the manuscript are very low resolution.

2. It would strengthen the findings of the manuscript if the authors confirm the results obtained using the innominate arteries by looking at total, HM / hybrid, a-2,6-sialylated, and a-2,3-sialylated ICAM-1 in the atherosclerotic mouse aorta.

3. How were the lesion and non-lesion segments identified?

4. Fig. 2 and 3 legends state that n=5 experiments were performed for CD68 immunostaining. The corresponding bar graphs show only 3 experiments.

5. What is the correlation coefficient for Fig. 4 studies? It seems there would be no correlation if only the closed symbols are taken into consideration. It is recommended to perform more experiments for the correlation studies.

In the high fat diet model (Fig. 2; no ligation), there was atherosclerosis in the brachiocephalic artery and HM/hybrid, HM, a-2,6-sialylated and a-2,3-sialylated ICAM-1 were increased in lesion vs. non-lesion segments. If there is atherosclerosis in these mice but no correlation between CD68 and ConA or HHL puncta, the mechanisms regulating ICAM N-glycosylation may not be important for monocyte adhesion and macrophage accumulation.

6. Type 2 human atherosclerotic lesions show no ICAM-1 expression. The type 2 lesions are characterized by the presence of lipid-laden macrophages according to Stary et al (ATVB, 1995). These findings suggest that ICAM-1 (and its N-glycoforms) does not play a role at the early phase of monocyte transmigration in human atherosclerotic vessels.

7. Fig. 6B legends n=10 is misleading. It seems only n=5 samples per group was involved in the analysis. The authors need to carefully check the legends for accuracy.

8. This is an observational study and no functional experiments have been included to investigate the role of different ICAM-1 N-Glycoforms in the development of atherosclerosis or pathogenesis of AVF maturation failure. The manuscript would be strengthened by the addition of mechanistic data.

9. The rational for including the AVF data in the manuscript is not entirely clear. The mechanisms, leading to endothelial dysfunction in AVF failure vs atherosclerosis is different.

Reviewer #2: This is a novel approach to assessment of ICAM-1 N-Glycoforms in mouse and human models of endothelial dysfunction. In this paper using the proximity ligation assay (PLA), authors assessed the relative formation of high mannose, hybrid and complex α-2,6-sialyated N-glycoforms of ICAM-1 in human and mouse models of atherosclerosis, as well as in arteriovenous fistulas (AVF) of patients on hemodialysis.

To establish the distinct ICAM1-N-glycoforms in inflammatory disease in ivvivo authors solely used PLA assay. Indeed this assay has some limitation, however the PLA positive red puncta was not clear in all images throughout the manuscript. As this data represents the first study to demonstrating ICAM1-N-glycoforms invivo authors should provide convincing images with good quality pictures for PLA assay.

There was no description for Figure 2F and 2G in result section. Author should mention about these two figures.

In result section, Figure 3A author described that total ICAM-1, HM / hybrid N-glycoforms were increased in the ligated left carotid artery (LC) compared to the control right carotid artery (RC). But this statement was not matches with the Figure 3A. It looks like total ICAM1 and α-2,6-sialylated ICAM1 expressed more than HM/hybride. Fig 3C quantification was not reflects the Fig 3A images. Author should be careful about this discrepancy.

In Fig 4 and Fig 6 show correlations between CD68 staining and HM / hybrid, HM and α-2,6- sialylated, ICAM-1 respectively. It was not clear how author made this correlation graph. Author should provide some direct evidence to prove this statement that CD68 expression higher in HM/hybrid not other ICAM1 N-glycoformes.

Partial ligation and HFD are well establish model for atherosclerosis. Authors used only male mice without justification for the exclusion of female mice. Author should comment on this in discussion.

Overall, this manuscript does not fit for publication in PLoS One as it currently stands.

Reviewer #3: The current study showed correlation of high mannose (HM)-ICAM1 N glycoforms and a-2,6 sialylated ICAM1 N-glycoforms with macrophage burden in the lesions in human and mouse atherosclerosis, and in AVF maturation failure in hemodialysis patients. HM-ICAM1 N glycofoms and sialylated ICAM1 N-glycoforms were assessed by proximity ligation assay, while macrophage burden were assessed by CD68 staining. This is an extension of author’s previous studies. Findings are novel and important. There are several issues that should be addressed to make manuscript stronger.

1) Overall, image quality except Figure 7A is poor. High resolution image will be needed, since this is a crux of current study.

2) To support HM-ICAM1 N glycoforms and a-2,6 sialylated ICAM1 N-glycoforms in lesions, they need to be shown by western analysis as well.

3) In panel 2F, 3D, 6A, please show macrophage staining by immunohistochemistry, instead of immunofluorescence.

---

## [Author Response · Author response to Decision Letter 0]

5 Feb 2020

Dear Dr. Fukai,

Thank you for forwarding reviewer comments on our article “Assessment of ICAM-1 N-glycoforms in Mouse and Human Models of Endothelial Dysfunction”. We appreciate the reviewer’s constructive comments. In the revised manuscript, we have addressed the comments raised. Specific responses are provided below with edits in the text indicated by red font color.

Thank you

Sincerely

Rakesh Patel, PhD

Reviewer #1:

The study by Regal-McDonald et al used PLA to investigate co-localization of specific N-glycan stuctures and ICAM-1 in mouse and human atherosclerotic arteries and human AVF of chronic kidney disease patients (models of endothelial dysfunction). The results demonstrated that hypoglycosylated and α-2,6-sialylated ICAM-1 N-glycoforms are present in atherosclerotic murine and human arteries and AVF patients. The results also indicated that HM-ICAM-1 levels positively correlated with macrophage burden in both murine and human atherosclerotic tissue. The study is timely but lacks functional information on the role of ICAM-1 N-glycoforms in atherosclerosis or AVF.

1. The figures of the manuscript are very low resolution. 

 Author response: Image resolution has been increased. Al images are at 600dpi resolution per journal guidelines.

2. It would strengthen the findings of the manuscript if the authors confirm the results obtained using the innominate arteries by looking at total, HM / hybrid, a-2,6-sialylated, and a-2,3-sialylated ICAM-1 in the atherosclerotic mouse aorta.

 Author response: We did not have access to mouse aortic tissue, but have now included new data from the left carotid arteries from ApoE-/- mouse fed high fa diet. These vessels are commonly used to interrogate atherosclerosis disease mechanisms; see revised Fig 2. 

3. How were the lesion and non-lesion segments identified?

Author response: Human lesions were characterized by a cardiovascular pathologist using the defining characteristics of fibrosis, calcification, and fatty deposits. Mouse lesion segments were identified by the presence of increased neointimal hyperplasia compared to non-lesion segments. 

4. Fig. 2 and 3 legends state that n=5 experiments were performed for CD68 immunostaining. The corresponding bar graphs show only 3 experiments.

Author response: We apologize for this oversight. The manuscript has been updated with additional experiments and n numbers corrected accordingly.

5. What is the correlation coefficient for Fig. 4 studies? It seems there would be no correlation if only the closed symbols are taken into consideration. It is recommended to perform more experiments for the correlation studies.

Author response: Additional replicates have been added and correlation analyses separated out into HFD-fed experiments and partial carotid ligation to promote clarity. 

6. In the high fat diet model (Fig. 2; no ligation), there was atherosclerosis in the brachiocephalic artery and HM/hybrid, HM, a-2,6-sialylated and a-2,3-sialylated ICAM-1 were increased in lesion vs. non-lesion segments. If there is atherosclerosis in these mice but no correlation between CD68 and ConA or HHL puncta, the mechanisms regulating ICAM N-glycosylation may not be important for monocyte adhesion and macrophage accumulation.

Author response: As explained above, additional experiments were performed. With these additional replicates and the separation of models in the correlation analysis, ConA and HHL had a significant positive correlation with CD68 staining in HFD-mice and partial carotid ligation.

7. Type 2 human atherosclerotic lesions show no ICAM-1 expression. The type 2 lesions are characterized by the presence of lipid-laden macrophages according to Stary et al (ATVB, 1995). These findings suggest that ICAM-1 (and its N-glycoforms) does not play a role at the early phase of monocyte transmigration in human atherosclerotic vessels.

Author Response: We do not make the claim that ICAM-1 N-glycoforms play a role in early monocyte adhesion. However, the lack of ICAM-1 staining may also be due in part to detection limitations in our assay. We have included discussion of this point in the revised manuscript. 

8. Fig. 6B legends n=10 is misleading. It seems only n=5 samples per group was involved in the analysis. The authors need to carefully check the legends for accuracy

Author response: Additional data was added to manuscript and legends adjusted accordingly.

9. This is an observational study and no functional experiments have been included to investigate the role of different ICAM-1 N-Glycoforms in the development of atherosclerosis or pathogenesis of AVF maturation failure. The manuscript would be strengthened by the addition of mechanistic data.

Author response: Recently published work out of our lab demonstrates that different ICAM-1 N-glycoforms selectively recruit monocyte subsets under flow. Please see citation # 18 in the manuscript.

10. The rational for including the AVF data in the manuscript is not entirely clear. The mechanisms, leading to endothelial dysfunction in AVF failure vs atherosclerosis is different.

Author response: Mechanisms leading to AVF do indeed differ from atherosclerosis, but endothelial dysfunction as a result of turbulent blood flow is an underlying component of both diseases states. We wanted to test distinct ICAM-1 N-glycoforms were also present in other disease settings where vascular inflammation and turbulent blood flow are underlying features. This is also further discussed in the manuscript. 

Reviewer #2: 

This is a novel approach to assessment of ICAM-1 N-Glycoforms in mouse and human models of endothelial dysfunction. In this paper using the proximity ligation assay (PLA), authors assessed the relative formation of high mannose, hybrid and complex α-2,6-sialyated N-glycoforms of ICAM-1 in human and mouse models of atherosclerosis, as well as in arteriovenous fistulas (AVF) of patients on hemodialysis.

1. To establish the distinct ICAM1-N-glycoforms in inflammatory disease in ivvivo authors solely used PLA assay. Indeed this assay has some limitation, however the PLA positive red puncta was not clear in all images throughout the manuscript. As this data represents the first study to demonstrating ICAM1-N-glycoforms invivo authors should provide convincing images with good quality pictures for PLA assay.

 Author response: Image resolution has been increased.

2. There was no description for Figure 2F and 2G in result section. Author should mention about these two figures.

Author response: We apologize for this oversight. The manuscript has been edited to include these figures in the results section.

3. In result section, Figure 3A author described that total ICAM-1, HM / hybrid N-glycoforms were increased in the ligated left carotid artery (LC) compared to the control right carotid artery (RC). But this statement was not matches with the Figure 3A. It looks like total ICAM1 and α-2,6-sialylated ICAM1 expressed more than HM/hybride. Fig 3C quantification was not reflects the Fig 3A images. Author should be careful about this discrepancy.

Author response: Analysis of PLA puncta is between paired LC and RC sections, not across lectins (i.e., α-2,6-sialylated ICAM-1 vs. HM/hybrid ICAM-1). Direct comparison between lectins is not possible as each have different binding affinities to their target epitopes- similar to how two different antibodies may bind with distinct affinities. The presented data is to show an increase in the ICAM-1 N-glycoforms in the ligated vessel compared to the non-ligated vessel (LC vs RC). We have clarified this point in the text. 

4. In Fig 4 and Fig 6 show correlations between CD68 staining and HM / hybrid, HM and α-2,6- sialylated, ICAM-1 respectively. It was not clear how author made this correlation graph. Author should provide some direct evidence to prove this statement that CD68 expression higher in HM/hybrid not other ICAM1 N-glycoformes.

Author response: Correlation studies were performed by measuring CD68 staining in the same (serial) sections on which PLA staining was performed. In the case of HM and hybrid ICAM-1, CD68 staining increased as puncta increased, indicating a positive association between the two. Statistical analyses were performed using Pearson correlation.

Reviewer #3: 

The current study showed correlation of high mannose (HM)-ICAM1 N glycoforms and a-2,6 sialylated ICAM1 N-glycoforms with macrophage burden in the lesions in human and mouse atherosclerosis, and in AVF maturation failure in hemodialysis patients. HM-ICAM1 N glycofoms and sialylated ICAM1 N-glycoforms were assessed by proximity ligation assay, while macrophage burden were assessed by CD68 staining. This is an extension of author’s previous studies. Findings are novel and important. There are several issues that should be addressed to make manuscript stronger.

1) Overall, image quality except Figure 7A is poor. High resolution image will be needed, since this is a crux of current study.

 Author response: Image resolution has been increased.

2) To support HM-ICAM1 N glycoforms and a-2,6 sialylated ICAM1 N-glycoforms in lesions, they need to be shown by western analysis as well.

 Author Response: By western blot analyses, we have shown previously that ICAM-1 immunoprecipitated from atherosclerotic lesions has a MWt close to 75KDa suggesting it is a HM-glycoform. Our published studies show that ICMA-1 with complex N-linked sugars has a MWt closer to 100KDa. This information is included in the revised discussion.

3) In panel 2F, 3D, 6A, please show macrophage staining by immunohistochemistry, instead of immunofluorescence.

 Author Response: Immunofluorescence has been used extensively to assess CD68 staining in lesions, and per our experience there are no pros/cons in IF vs IHC for this marker. We have added an IgG staining control to Figure 6 to further verify specificity of the antibodies used for IF-based detection of CD68.

---

## [Decision Letter · Decision Letter 1]

28 Feb 2020

Assessment of ICAM-1 N-Glycoforms in Mouse and Human Models of Endothelial Dysfunction

PONE-D-19-20764R1

Dear Dr. Rakesh Patel,

We are pleased to inform you that your manuscript has been judged scientifically suitable for publication and will be formally accepted for publication once it complies with all outstanding technical requirements.

With kind regards,

Tohru Fukai

Academic Editor

PLOS ONE

Additional Editor Comments (optional):

Reviewers' comments:

Reviewer's Responses to Questions

**Comments to the Author**

1. If the authors have adequately addressed your comments raised in a previous round of review and you feel that this manuscript is now acceptable for publication, you may indicate that here to bypass the “Comments to the Author” section, enter your conflict of interest statement in the “Confidential to Editor” section, and submit your "Accept" recommendation.

Reviewer #1: All comments have been addressed

Reviewer #3: All comments have been addressed

2. Is the manuscript technically sound, and do the data support the conclusions?

Reviewer #1: Yes

Reviewer #3: Yes

3. Has the statistical analysis been performed appropriately and rigorously? 

Reviewer #1: Yes

Reviewer #3: Yes

4. Have the authors made all data underlying the findings in their manuscript fully available?

Reviewer #1: Yes

Reviewer #3: Yes

5. Is the manuscript presented in an intelligible fashion and written in standard English?

Reviewer #1: Yes

Reviewer #3: Yes

6. Review Comments to the Author

Reviewer #1: (No Response)

Reviewer #3: Revised manuscript has been improved, since image quality has been improved and necessary information was added. No further comments.

7. PLOS authors have the option to publish the peer review history of their article (what does this mean?). If published, this will include your full peer review and any attached files.

Reviewer #1: No

Reviewer #3: No

---

## [Editor Report · Acceptance letter]

3 Mar 2020

PONE-D-19-20764R1 

Assessment of ICAM-1 N-Glycoforms in Mouse and Human Models of Endothelial Dysfunction 

Dear Dr. Patel:

I am pleased to inform you that your manuscript has been deemed suitable for publication in PLOS ONE. Congratulations! Your manuscript is now with our production department. 

With kind regards,

on behalf of

Dr. Tohru Fukai 

Academic Editor

PLOS ONE